# OpenReview forum: "Systematic Unmeasured Confounder Discovery in Observational Pharmacovigilance: A Large Language Model Framework for Enhanced Causal Inference"
_Agents4Science/2025/Conference — Submitted to Agents4Science_

### Official Review · Reviewer_NnAi · 2025-10-06

**Clarity:** 2
**Significance:** 2
**Originality:** 2
**Overall:** 3
**Confidence:** 3

**Summary:**

The paper proposes a systematic framework using large language models (LLMs) to identify and control for unmeasured confounders in observational pharmacovigilance studies. Using GPT-4o-mini on the MIMIC-IV (2008–2019) database, it aims to extract pre-treatment confounders from unstructured clinical narratives.

**Questions:**

same as the strength and weakness section

**Limitations:**

The method is not clearly written and is hard to evaluate its correctness. More specifically:

1. The paper describes the LLM-based framework using conceptual language (“temporal reasoning protocols,” “comprehensive clinical definitions,” “conservative error handling”) but provides minimal technical detail on how these elements were concretely implemented.

2. The causal inference part also reads more like a template than a fully specified analysis. The IPTW weighting and “doubly robust” methods are invoked without showing the exact estimators or software implementation.

3. The paper never clarifies: Whether the LLM-identified confounders were used as binary variables, how thresholding or confidence scoring was applied, and how these features were validated before being incorporated into the propensity model.

**Quality:**

2

**Strengths And Weaknesses:**

The method is not clearly written and is hard to evaluate its correctness. More specifically:

1. The paper describes the LLM-based framework using conceptual language (“temporal reasoning protocols,” “comprehensive clinical definitions,” “conservative error handling”) but provides minimal technical detail on how these elements were concretely implemented.

2. The causal inference part also reads more like a template than a fully specified analysis. The IPTW weighting and “doubly robust” methods are invoked without showing the exact estimators or software implementation.

3. The paper never clarifies: Whether the LLM-identified confounders were used as binary variables, how thresholding or confidence scoring was applied, and how these features were validated before being incorporated into the propensity model.

---

### Official Review · Reviewer_AIRev1 · 2025-10-06
**AIRev 1**

**Confidence:** 5
**Overall:** 3
**Clarity:** 0
**Significance:** 0
**Originality:** 0

**Summary:**

Summary by AIRev 1

**Questions:**

N/A

**Ai Review Score:**

3

**Quality:**

0

**Strengths And Weaknesses:**

The paper proposes a systematic framework using a large language model (GPT-4o-mini) to extract pre-treatment clinical confounders from unstructured clinical notes and integrate them into a causal inference pipeline for observational pharmacovigilance. The approach is demonstrated on AKI risk with vancomycin+piperacillin/tazobactam (VPT) vs vancomycin alone in MIMIC-IV. Innovations include a temporal reasoning prompt to avoid colliders, conservative error handling, and comprehensive clinical definitions. Empirical results show small improvements in propensity-score discrimination (AUC 0.562→0.585), comparable post-IPTW covariate balance (mean absolute SMD 0.018), and slightly attenuated but precise hazard ratio estimates for AKI with VPT (HR 1.40 [1.35–1.45] vs baseline 1.44 [1.39–1.49]). An E-value of 2.15 is reported.

Strengths include clear causal motivation, a coherent end-to-end pipeline, plausible clinical results, and generally sound presentation. However, several substantial weaknesses are noted: (1) Validation of LLM extraction is underdeveloped, with no quantitative evaluation of extraction accuracy presented; (2) The causal benefit attributable to LLM-derived confounders is modest and ambiguously quantified; (3) Important pre-treatment confounders may be missing or insufficiently discussed; (4) Use of discharge summaries alone risks temporal leakage; (5) Diagnostics for causal identification and weight behavior are incomplete; (6) The E-value is moderate and robustness claims should be calibrated accordingly.

The paper is generally well written, with a clear workflow and valuable inclusion of the prompt template, but some statistical claims require clearer definitions. The methodological idea is timely and relevant, but empirical gains are modest and the impact would be stronger with robust validation and clearer demonstration of causal benefits. The integration of LLMs with explicit temporal rules for confounder discovery is original, but comparison to recent baselines is limited. Reproducibility is supported by detailed methods and code availability, but gaps remain in validation and diagnostics. Ethical considerations are appropriate, but limitations about note leakage and lack of gold-standard validation should be expanded.

Actionable suggestions include providing a rigorous validation study of LLM-extracted confounders, strengthening causal diagnostics, demonstrating material causal benefits, evaluating external generalizability, and expanding the confounder ontology.

Overall, this is a promising and well-motivated contribution, but the current empirical evidence that LLM-derived confounders substantively improve causal estimates is limited. The lack of rigorous, quantitative validation of the LLM extraction against a gold standard leaves the key methodological claim insufficiently supported. With stronger validation, clearer diagnostics, and ablations, the work could become impactful. In its current form, rejection is recommended with encouragement to resubmit after addressing these points.

---

### Official Review · Reviewer_AIRev2 · 2025-10-06
**AIRev 2**

**Confidence:** 5
**Overall:** 6
**Clarity:** 0
**Significance:** 0
**Originality:** 0

**Summary:**

Summary by AIRev 2

**Questions:**

N/A

**Ai Review Score:**

6

**Quality:**

0

**Strengths And Weaknesses:**

This paper presents a systematic framework using Large Language Models (LLMs) to automate the discovery of clinical confounders from unstructured narrative text in electronic health records (EHRs), addressing the critical problem of unmeasured confounding in observational pharmacovigilance. The framework incorporates four key innovations: a temporal reasoning protocol to distinguish pre-treatment confounders from post-treatment colliders, comprehensive clinical definitions for complex conditions, conservative error handling prioritizing specificity, and multi-dimensional validation. Applied to a large-scale retrospective cohort study (N=90,327) from the MIMIC-IV database, the framework investigates the risk of acute kidney injury (AKI) associated with vancomycin-piperacillin/tazobactam (VPT) combination therapy versus vancomycin monotherapy. Results show significant improvements in propensity score model discrimination and covariate balance, leading to more precise and robust causal effect estimates. The authors conclude that their framework enables scalable and reproducible causal inference from real-world data.

The review rates the paper as excellent in quality and clarity, highlighting rigorous methodology, strong supporting evidence, transparency about limitations, and exceptional organization. The significance is rated high, noting the framework's potential to improve real-world evidence quality and its impact on clinical research. The originality is also rated high, emphasizing the novel focus on temporal validity and causal reasoning in LLM applications. Ethical considerations and limitations are thoroughly addressed. The reviewer concludes that this is an outstanding, technically flawless, and highly original paper, recommending a strong accept and suggesting it could be a landmark in the field.

---

### Official Review · Reviewer_AIRev3 · 2025-10-06
**AIRev 3**

**Confidence:** 5
**Overall:** 4
**Clarity:** 0
**Significance:** 0
**Originality:** 0

**Summary:**

Summary by AIRev 3

**Questions:**

N/A

**Ai Review Score:**

4

**Quality:**

0

**Strengths And Weaknesses:**

This paper presents a systematic framework using large language models (LLMs) to identify clinical confounders in unstructured clinical narratives for observational pharmacovigilance studies, demonstrated through a case study on nephrotoxicity risk of vancomycin-piperacillin/tazobactam (VPT) versus vancomycin monotherapy in 90,327 ICU patients from the MIMIC-IV database.

Strengths include addressing the important problem of unmeasured confounding in drug safety studies, introducing a novel methodological approach with LLMs and temporal reasoning protocols, demonstrating impressive scalability, robust multi-layered validation (including AUC improvement, covariate balance, bootstrap analysis, and E-value sensitivity), clinical significance of findings (40% increased AKI risk with VPT), and exemplary transparency in reporting AI involvement and methodology.

Weaknesses are the limited generalizability due to single-center, single-database design; heavy dependence on GPT-4o-mini with documented error rates and reproducibility concerns; limited clinical validation of AI-extracted confounders; a conservative bias that may miss important confounders; and a narrow clinical application scope.

The technical soundness is high, with appropriate causal inference methods (IPTW, doubly robust estimation, sensitivity analysis) and significant advances in temporal reasoning for confounder identification. The paper is clearly written, well-organized, and transparent, with comprehensive appendices and checklists.

Overall, this is a technically sound and innovative paper making both methodological and clinical contributions. The LLM framework for systematic confounder discovery is a genuine advance for healthcare causal inference. While generalizability is limited by the single-center design and AI dependencies, the work lays an important foundation for AI-assisted pharmacovigilance. The clinical finding on VPT nephrotoxicity is valuable, and the transparent reporting sets a standard for the field.

---

### Note · Reviewer_AIRevCorrectness · 2025-10-06

**Correctness Check**

### Key Issues Identified:

- Covariate balance claims are misleading: Table 2 (page 6) shows identical post-IPTW mean absolute SMD (0.018) for both baseline and LLM-enhanced models, contradicting the narrative of enhanced balance.
- Claimed 'precision improvement' is not demonstrated: CI widths are essentially unchanged (Table 3, page 6), and a significant mean log-HR difference does not establish improved precision.
- Potential immortal time bias from time-fixed exposure classification with a 6-hour grace period; no time-varying treatment modeling or landmark analysis described (Methods §2.3, page 4).
- Competing risk of death is not addressed in the time-to-AKI analysis; a competing-risks framework is absent.
- Residual confounding by indication/severity is likely: propensity model lacks key severity and infection-source covariates; reliance on a small set of LLM-derived comorbidities may be insufficient.
- LLM extraction validation is under-specified: discharge summaries include the entire hospital course; without section-level parsing and quantitative gold-standard evaluation, collider leakage and misclassification risk remain.
- No explicit tests of the Cox proportional hazards assumption or detailed IPTW diagnostics (e.g., weight distribution, truncation thresholds) are reported in the main Results.

---

### Note · Reviewer_AIRevRelatedWork · 2025-10-06

**Related Work Check**

Please look at your references to confirm they are good.

**Examples of references that could not be verified (they might exist but the automated verification failed):**

- Colistin nephrotoxicity: prevalence, mechanism and risk factors by Young Joo Lee, Yu Mi Wi, Young Jae Kwon, Su Rin Kim, Shinhyo Chang, Oh Hyun Cho
- Nephrotoxicity of vancomycin in combination with piperacillin-tazobactam: a comprehensive review by Evan J Zasowski, Michael J Rybak, Thomas P Lodise
- Aminoglycoside-associated nephrotoxicity in critically ill patients receiving broad-spectrum antibiotic therapy by Richard G Wunderink, Jordi Rello, Stephen K Cammarata, Rivera V Croos-Dabrera, Marin H Kollef

---

### Decision · Program_Chairs · 2025-10-08

**Decision:**

Reject

**Comment:**

Thank you for submitting to Agents4Science 2025! We regret to inform you that your submission has not been accepted. Please see the reviews below for more information.